# Factors That Affect the Enlargement of Bacterial Protoplasts and Spheroplasts

**DOI:** 10.3390/ijms21197131

**Published:** 2020-09-27

**Authors:** Hiromi Nishida

**Affiliations:** Department of Biotechnology, Toyama Prefectural University, 5180 Kurokawa, Imizu, Toyama 939-0398, Japan; hnishida@pu-toyama.ac.jp; Tel.: +81-766-56-7500

**Keywords:** bacterial cell enlargement, microinjection, protoplasts, spheroplasts, metal ion composition, DNA replication, vacuole formation

## Abstract

Cell enlargement is essential for the microinjection of various substances into bacterial cells. The cell wall (peptidoglycan) inhibits cell enlargement. Thus, bacterial protoplasts/spheroplasts are used for enlargement because they lack cell wall. Though bacterial species that are capable of gene manipulation are limited, procedure for bacterial cell enlargement does not involve any gene manipulation technique. In order to prevent cell wall resynthesis during enlargement of protoplasts/spheroplasts, incubation media are supplemented with inhibitors of peptidoglycan biosynthesis such as penicillin. Moreover, metal ion composition in the incubation medium affects the properties of the plasma membrane. Therefore, in order to generate enlarged cells that are suitable for microinjection, metal ion composition in the medium should be considered. Experiment of bacterial protoplast or spheroplast enlargement is useful for studies on bacterial plasma membrane biosynthesis. In this paper, we have summarized the factors that influence bacterial cell enlargement.

## 1. Introduction

Bacteria usually grow asexually, inheriting their DNA by clonal production (cell division). Therefore, these clones have the same genetic information. If clonal production were the only form of inheritance in bacteria, bacteria would display limited genetic variation. However, bacteria are so diverse that more than 99% are thought to be unknown at the species level. The horizontal transfer of genetic information between heterologous bacteria greatly contributes to genetic variation [1,2,3,4,5,6,7].

Which genetic elements have been maintained or lost during bacterial evolution? Has the acceptance of genetic information occurred randomly? Thus far, these remain open questions in the field of molecular biology. This is because, by using gene manipulation technologies, experiments have been done in specific genes only or in a limited number of genes, making it impossible to investigate the system of bacterial genome evolution. In contrast, the genome manipulation technology that has been recently reported is based on the repetition of existing gene manipulation technologies [8,9,10,11,12]. Unfortunately, the number of bacterial species capable of gene manipulation, such as *Bacillus subtilis* and *Mycoplasma mycoides*, used in this kind of experiments, is still limited. In addition, these experiments are time-consuming, labor-intensive, and costly. Thus, it is difficult to perform them in regular laboratories.

For genomic sciences, changing from gene manipulation technologies to genome manipulation technology, in which a whole genome may be treated as if it were one gene, could be very useful. For example, for the introduction of *B. subtilis* genomic DNA into *Escherichia coli* cells at one time, to be able to examine how many genes are expressed from the heterologous genome in such a system. Currently, there is no way to introduce whole heterologous genomic DNA into bacterial host cells at once. 

In this review, the bacterial cells lacking cell wall with an outer membrane and a plasma membrane are called spheroplasts, and those without an outer membrane are called protoplasts. Thus, protoplasts and spheroplasts are produced from Gram-positive and Gram-negative bacteria, respectively. Cell wall (peptidoglycan) biosynthesis plays a crucial role in bacterial cell shape maintenance. Bacterial cells cannot enlarge in the presence of an intact peptidoglycan sacculus. Thus, it is necessary to stop peptidoglycan biosynthesis in order to enlarge bacterial cells. On the other hand, cell wall is not essential for bacteria to survive because many bacteria can change to L-form bacteria that are capable of dividing, increasing the number of cells without cell wall. L-form bacteria have been detected and isolated from various environments and most of them are antibiotic-resistant [13,14,15,16,17,18]. The spheroplast incubation method is used to enlarge bacterial cells [19]. Bacterial protoplasts/spheroplasts are produced by lysing the cell wall with lysozyme or by penicillin [20]. In the spheroplast incubation method, protoplasts/spheroplasts are produced by lysozyme. Though bacterial protoplasts/spheroplasts cannot divide, they may enlarge under suitable culture conditions where cell wall synthesis is inhibited (Figure 1). Enlarged bacterial cells have already been used in patch clamp analyses [19,21,22,23], but only recently, a microinjection method has been established for these enlarged cells [24]. Noteworthy, different bacterial species have different patterns of cell enlargement. 

Here, we summarize the factors that influence the enlargement of bacterial cells, based on available literature. In our laboratory, the following bacterial cells have been enlarged: *Bacillus subtilis*, *Deinococcus grandis*, *Enterobacter hormaechei*, *Enterococcus faecalis*, *Erythrobacter litoralis*, *Escherichia coli*, *Lelliottia amnigena*, and *Rhodospirillum rubrum*. The native forms of these 8 species are a few micrometers in diameter. Among them, *B. subtilis* and *E. faecalis* are Gram-positive. The other bacteria are Gram-negative. Commonality and diversity are observed in bacterial protoplast or spheroplast enlargement. We chose to work with protoplasts/spheroplasts as cells that could not divide, which differ from L-form bacteria. L-form bacteria have been detected from various environments because they can divide. In contrast, it has been exceedingly difficult to detect bacterial protoplasts/spheroplasts in nature because they do not grow. More works are needed to elucidate functions for bacterial protoplast or spheroplast formation in nature. 

## 2. Osmotic Pressure

The osmotic pressure of incubation media plays an important role in the maintenance of bacterial protoplasts/spheroplasts [25]. They do not enlarge when under an osmotic pressure higher than the suitable one [26], whereas their plasma membranes break when the osmotic pressure is below the optimum. For example, in spheroplasts of the Gram-negative radiation-resistant bacterium *Deinococcus grandis*, the plasma membrane breaks under a low osmotic pressure, whereas the outer membrane does not [26]. Thus, *D. grandis* cells seem to be maintained under low osmotic pressure; however, plasma membrane-broken cells are dead and cannot enlarge [26]. In *D. grandis* spheroplasts, the outer membrane has a higher osmotic pressure resistance than the inner (plasma) membrane. This may be related to the fact that both the cell wall and the outer membrane have an important role in cell shape maintenance [27].

We used Difco Marine Broth (5 g/L peptone, 1 g/L yeast extract, 0.1 g/L ferric citrate, 19.45 g/L NaCl, 5.9 g/L MgCl_2_, 3.24 g/L MgSO_4_, 1.8 g/L CaCl_2_, 0.55 g/L KCl, 0.16 g/L NaHCO_3_, 0.08 g/L KBr, 34 mg/L SrCl_2_, 22 mg/L H_3_BO_3_, 8 mg/L Na_2_HPO_4_, 4 mg/L Na_2_SiO_3_, 2.4 mg/L NaF, and 1.6 mg/L NH_4_NO_3_; BD, Franklin Lakes, NJ) containing penicillin (DMBp) as an incubation medium for cell enlargement of both marine and non-marine bacteria. We chose this medium because it has a higher osmotic pressure than other media and because most bacterial protoplasts/spheroplasts live stably under seawater osmotic conditions [24,26,28].

During cell enlargement, the lipid composition of the plasma membrane changes [28], indicating that the properties of the membrane differ between bacterial cells capable of dividing and enlarged protoplasts/spheroplasts. Bacterial protoplast or spheroplast enlargement is not accompanied by swelling due to water invasion into the cells, but it is accompanied by plasma (and outer) membrane biosynthesis [24,28]. More studies are needed to elucidate whether osmotic pressure affects lipid composition. When using metal salts or sugars as osmotic stabilizers, it should be noted that the function is often not limited to osmotic pressure adjustment (see below).

## 3. Metal Ions

Tryptone glucose yeast extract (TGY: 5 g/L tryptone, 1 g/L glucose, and 3.0 g/L yeast extract) medium is commonly used for culturing *Deinococcus*. *D. grandis* spheroplast enlargement was not observed in TGY plus penicillin medium containing sucrose as an osmotic stabilizer, but it was observed when the medium was supplemented with metal salts instead [28]. In order to investigate which metal salt plays an important role in spheroplast enlargement, we compared the cell morphology among *D. grandis* spheroplasts in incubation media with different metal salt compositions. We determined that either calcium or magnesium ions were essential for *D. grandis* spheroplast enlargement [28]. 

It is known that calcium and magnesium ions bind lipopolysaccharides (LPS) on the outer membrane of Gram-negative bacteria to maintain the cell surface structure [29,30,31]. However, *Deinococcus* has a unique lipid composition [32,33] and no LPS [34], suggesting that *D. grandis* may have another component that binds calcium/magnesium ions on its cell surface. In *E. coli*, calcium and magnesium ions play a role in lipid biosynthesis [35,36]. It suggests that calcium/magnesium ions may also play a role in lipid biosynthesis in *D. grandis*. 

Interestingly, outer membrane fusion occurs when incubation is done in a medium containing calcium ions [26,37], but not when incubated in a medium containing magnesium ions. Outer membrane fusion was observed at 200 mM of calcium ions, but was not observed at 100 mM of both calcium and magnesium ions as well as at 200 mM of magnesium ions [26]. These results indicate that calcium and magnesium ions do not have the same function in the enlargement of the outer membrane of *D. grandis* [26,37]. 

During *D. grandis* spheroplast enlargement, the synthesis of the outer membrane is much faster than that of the inner membrane in the presence of calcium ions [26,28,37]. This strongly suggests that the outer membrane has a calcium ion-binding component associated with its expansion and fusion.

*Deinococcus* has surface (S)-layer proteins in its cell surface [38,39]. S-layer is two-dimensional protein array that are frequently found on the bacterial and archaeal surfaces [40]. *D. radiodurans* has two major S-layer proteins, Hpi and SlpA [39]. Hpi locates on outer membrane, and SlpA binds Hpi on one side and peptidoglycan on the other side [39]. Thus, we constructed an S-layer protein coding gene (*slpA*) deletion mutant of *D. grandis*. These mutant’s spheroplasts enlarged as those from the wild type strain (unpublished data), suggesting that the SlpA homologous protein may not be the calcium ion-binding component that we were searching for.

*D. grandis* is rod-shaped and has a *rodZ* gene, a homolog to that of *E. coli* [41]. *E. coli* RodZ is a transmembrane protein that binds MreB, a cell shape-determining protein, inside the plasma membrane [42,43,44,45]. A *D. grandis rodZ* deletion mutant has a higher sensitivity to calcium ion than wild type strains [46]. Thus, at low calcium ion concentrations, where wild type spheroplasts cannot enlarge, *rodZ*-deletion mutants can [46]. The mechanism underlying increased sensitivity for calcium ion by the *rodZ*-deletion spheroplasts of *D. grandis* needs further investigation. However, this result indicates that calcium ion sensitivity is associated with spheroplast enlargement. Interestingly, an *mreB* deletion mutant in the Gram-positive bacterium *B. subtilis* requires magnesium ion for normal growth [47]. These findings suggest that the MreB-RodZ system for bacterial cell shape maintenance may be associated with calcium or magnesium ions. 

Enlarged protoplasts of the Gram-positive lactic acid bacterium *Enterococcus faecalis* incubated in DMBp are useful for microinjection [24]. However, for the Gram-negative bacterium *Lelliottia amnigena* spheroplasts, the success rate of microinjection is very low when incubation is done in DMBp [24]. However, after cultivating these spheroplasts in media supplemented with different metal salt compositions, it was determined that the optimum metal salt composition for their enlargement was 62 mM CaCl_2_, 7.4 mM KCl, and 16.2 mM MgCl_2_ [24]. This metal salt composition was more suitable for enlargement than the composition of 16.2 mM CaCl_2_, 7.4 mM KCl, and 62 mM MgCl_2_ [24], which is the same concentration of CaCl_2_, KCl, and MgCl_2_ in DMB. This indicates that the plasma membrane properties depend on the metal salt composition in the incubation medium. 

The response to metal ions differs among different bacterial species. For example, the magnesium ion influx-related gene *corA* is upregulated in enlarged spheroplasts of the photosynthetic bacterium *Erythrobacter litoralis*, but it is downregulated in *L. amnigena* enlarged spheroplasts [48]. In contrast, the magnesium ion efflux-related gene *corD* is downregulated in *E. litoralis,* but upregulated in *L. amnigena* [48]. This suggests that *E. litoralis* enlarged spheroplasts need magnesium ions for their maintenance, but *L. amnigena* spheroplasts do not. Transcriptome and proteome analyses are useful for studies on the relationship between metal ions and cell enlargement. Based on our previous studies, it appears that there is no metal ion with common functions in all bacterial protoplasts/spheroplasts.

## 4. Photosynthesis and Respiration

The aerobic anoxygenic photosynthetic bacterium *E. litoralis* was used to investigate how photosynthesis and respiration were involved in spheroplast enlargement. *E. litoralis* spheroplasts enlarge in DMBp under dark conditions in the presence of oxygen, indicating that their respiration mechanisms are active during cell enlargement [49,50]. In addition, they enlarge under light/dark conditions in the absence of oxygen, indicating that their photosynthesis is also active during cell enlargement [50]. *Erythrobacter* synthesizes ATP through photosynthesis under aerobic conditions, but not under anaerobic conditions [51]. Thus, spheroplasts have different photosynthetic characteristics than intact cells. They do not enlarge when cultured under anaerobic and dark conditions, indicating that either respiration or photosynthesis is essential for cell enlargement [50].

Continuous light or blue light inhibits *E. litoralis* spheroplast enlargement [49,52]. Considering that the photosynthetic apparatus is generally synthesized in the dark [53,54], the enlarged spheroplasts of *E. litoralis* may also generate their photosynthetic apparatuses in the dark. It is known that blue light represses photosynthesis gene expression in the photosynthetic bacterium *Rhodobacter sphaeroides* [55]. Thus, blue light may also inhibit photosynthesis gene expression in *E. litoralis* spheroplasts [56]. *E. litoralis* spheroplasts do not enlarge under blue light but enlarge under green and red lights, indicating that they respond to different light signals [52]. These results support the notion that *E. litoralis* spheroplasts have an active photosynthetic apparatus. 

Not only *E. litoralis* spheroplasts but also other bacterial protoplasts/spheroplasts enlarge in the presence of oxygen. On the other hand, aerobic culture conditions inhibit L-form bacterial growth [57,58]. Thus, the response to oxygen differs between bacterial protoplasts/spheroplasts and L-form cells. Although L-form bacteria also lack cell walls, their cells can divide [15,58]. Further investigation is needed to elucidate how oxygen affects cell division and cell enlargement of cell wall-lacking bacteria. 

## 5. Chromosomal DNA (Replication)

Gene expression patterns differ between cells with normal division and enlarged bacterial protoplasts/spheroplasts [48,59]. In addition, both transcription and translation are essential for bacterial protoplast or spheroplast enlargement because inhibitors of transcription and translation efficiently prevent *D. grandis* spheroplast enlargement [28]; however, it is uncertain whether DNA replication is also essential for this enlargement. DNA replication occurs generally before cell division in all living organisms. Bacteria have several checkpoints along the cell division cycle [60,61,62,63,64,65,66,67,68]. Bacterial protoplasts/spheroplasts cannot divide but are enlarged in the presence of inhibitors of peptidoglycan synthesis [69]. To recover to their native forms, cell wall resynthesis is necessary [70,71]. Although bacterial protoplasts/spheroplasts cannot divide, DNA replication occurs during cell enlargement [19,21,49,72]. In the presence of the peptidoglycan biosynthetic inhibitor penicillin, *E. coli* forms a bulge at the site where the new cell wall is normally formed, which requires DNA replication [73]. These results indicate that DNA replication affects plasma membrane synthesis.

In the protoplasts of *E. faecalis*, the amount of DNA increases during the protoplast enlargement between 0 and 120 h of incubation in DMBp [74]. Both cell enlargement and DNA replication stop at 120 h of incubation, with no further change in cell size and DNA amount [74]. In order to investigate whether DNA replication influences bacterial protoplast or spheroplast enlargement, the DNA replication inhibitor novobiocin [75,76] was added to the incubation medium during the enlargement of *E. faecalis* protoplasts [74,77]. As a result, novobiocin inhibited not only DNA replication but also cell enlargement [74,77]. Although mitomycin C degrades chromosomal DNA of *E. faecalis* protoplasts, novobiocin does not [74]. When novobiocin was removed from the medium, *E. faecalis* protoplasts enlarged again [74,77]. Thus, DNA replication is associated with the biosynthesis of the plasma membrane. Bacterial protoplast/spheroplast enlargement can be controlled using novobiocin. 

Bacterial chromosomal DNA attaches to the plasma membrane for an accurate distribution during cell division [78,79,80,81,82,83]. However, it is uncertain whether chromosomal DNA attaches to the plasma membrane of enlarged protoplasts/spheroplasts, whether this attachment is required for enlargement; and whether the DNA replication system is associated with plasma membrane biosynthesis in the cytoplasm. Therefore, further studies are needed to further understand these matters. 

Interestingly, bacteria with large cell sizes have multiple copies of the genome in a single cell [84,85,86]. In *B. subtilis* and the large bacterium *Epulopiscium*, cell size and copy number of the genome have been shown to be positively correlated [84,87].

## 6. Vacuole Formation

In this review, “vacuole” is used for “provacuole” or “vacuole-like structure” in previous studies [19,21]. The size of *E. litoralis* enlarged spheroplasts is limited to a diameter of 6–7 μm; characteristically, they lack vacuoles [48,49]. In contrast, other bacterial protoplasts/spheroplasts exceed 15 μm in diameter and have vacuoles in their cytoplasm (Figure 1) [19,21,24]. Thus, vacuole generation and growth occur in the cytoplasm after plasma membrane expansion. If these vacuoles were not generated, cell enlargement would stop. Noteworthy, vacuole characteristics differ among bacterial species. For example, among the bacterial protoplasts/spheroplasts used in our laboratory, those from *L. amnigena* produce more vacuoles than any other, whereas those from *E. faecalis* produce the largest (Figure 2) [24]. It is known that the biosynthesis of the vacuole membrane continues during cell enlargement [24]. Thus, vacuole generation and expansion may be essential for bacterial protoplast or spheroplast enlargement over 15 μm in diameter. In other words, plasma membrane and vacuole membrane syntheses may be interrelated. 

Transmission electron microscopy images show that enlarged *E. faecalis* protoplasts bear discoidal structures in the cytoplasm; structures that have not been observed in any other bacterial protoplast/spheroplast (Figure 2) [24]. These structures contain no DNA or vacuoles [24]. Thus, more studies are needed to elucidate whether they are associated with vacuoles in *E. faecalis* protoplasts. 

Mature vacuoles are not connected to the plasma membrane [24]. However, the vacuolar membrane has proteins that are similar to those of the plasma membrane, such as respiratory chain proteins, F_0_F_1_-ATPase, and penicillin binding proteins [19,21,24]. This indicates that the vacuolar membrane components are the same or, at least, they are very similar to those of the plasma membrane. In addition, vacuoles of *E. coli* and *B. subtilis* enlarged protoplasts/spheroplasts have everted membranes, strongly suggesting that the vacuolar membrane may be generated by endocytosis of the plasma membrane [19,21]. 

After the DNA replication inhibitor novobiocin is introduced prior to vacuole formation during the enlargement of the *E. faecalis* protoplasts, the cell size is limited to 6 μm in diameter and the cells lack vacuoles [74]. It indicates that plasma membrane biosynthesis and vacuole formation require DNA replication in *E. faecalis* protoplasts. 

Interestingly, bacteria with large cell sizes also generate vacuoles in their cytoplasm [86,88,89]. For example, the largest bacterium known, *Thiomargarita namibiensis,* has a large vacuole [88,89] in which nitrate accumulates. Given that sulfur is present in the cytoplasm, away from nitrate, it is thought that the *T. namibiensis* vacuole has a different function than that of vacuoles from enlarged protoplasts/spheroplasts. 

## 7. Conclusions

Bacterial protoplasts/spheroplasts enlarge in the incubation medium containing an inhibitor of peptidoglycan synthesis under osmotically protective conditions. 

Plasma membrane biosynthesis plays a central role in bacterial protoplast or spheroplast enlargement. Metal ions, especially calcium and magnesium ions, affect the properties of plasma membranes of bacterial protoplasts/spheroplasts such as flexibility suggesting that metal ion composition in the incubation medium may influence the final lipid composition of the plasma membrane of these cells. In order to generate bacterial protoplasts/spheroplasts suitable for microinjection, it is important to consider the metal salt composition in the incubation medium. 

Plasma membrane biosynthesis (cell enlargement) requires ATP, which may be generated either during respiration or photosynthesis. Thus, it is important that respiration or photosynthesis systems are maintained in enlarged plasma membranes of bacterial protoplasts/spheroplasts. Plasma membrane expansion induces an increase in cytoplasmic volume. Thus, as protoplasts/spheroplasts become larger, ATP production in the plasma membrane alone becomes insufficient for maintaining the cytoplasm. 

Besides, during plasma membrane expansion, DNA replication occurs, and vacuoles are generated and enlarged in the cytoplasm. Generation and enlargement of vacuoles in the cytoplasm repress cytoplasmic volume increase. In addition, an increase in chromosomal copy number may activate transcription in the cytoplasm. Thus, plasma membrane biosynthesis in bacterial protoplasts/spheroplasts is associated with DNA replication and vacuole formation. More work is needed to elucidate the relationships among plasma membrane biosynthesis, DNA replication, and vacuole formation at the molecular level.

## Figures and Tables

**Figure 1 ijms-21-07131-f001:**
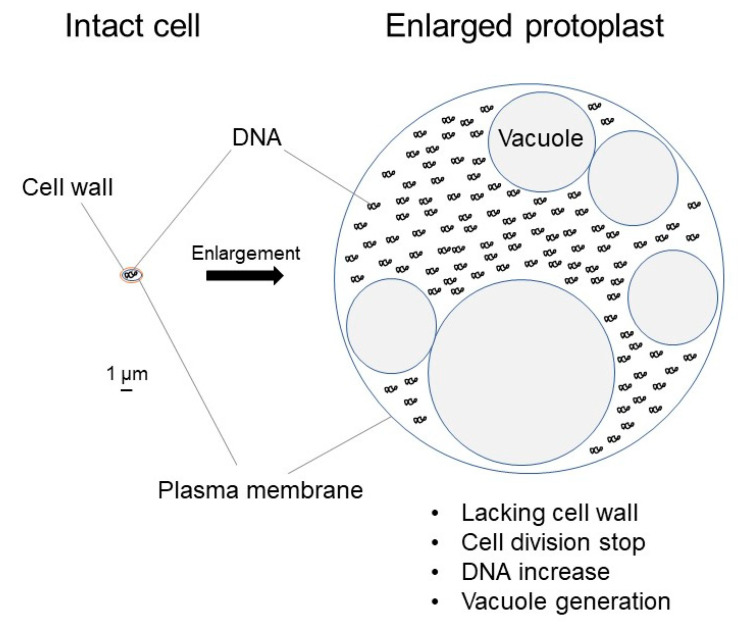
Comparison between intact bacterial cell and enlarged protoplast.

**Figure 2 ijms-21-07131-f002:**
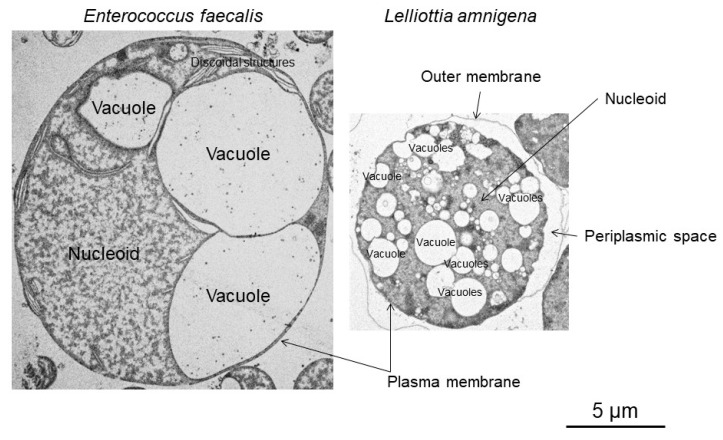
Transmission electron microscopy images of Gram-positive *Enterococcus faecalis* and Gram-negative *Lelliottia amnigena*.

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
