# Peer review of "Factors That Affect the Enlargement of Bacterial Protoplasts and Spheroplasts"

_ijms, 2020, doi:10.3390/ijms21197131_

Round 1

Reviewer 1 Report

At the moment, the manuscript reads like more of a promotion for the Author’s lab than for an objective global review of the topic at hand.  This is particularly acute in the Abstract (“our targets”, “our procedure”, and “our experimental method”).

Are there different ways to make protoplasts/spheroplasts? Mechanical vs enzymatic?  What are the benefits/drawbacks?  If one way is “better” than another, then describe why.

As presented here, the Author describes various characteristics of protoplasts/spherplasts.  However, he also mentions the possibility that these structures are formed in the environment.  What would be potential roles/functions for protoplast/spheroplast formation by bacteria in nature? (i.e. what are they used for?)

Given the extensive literature and work by the author in this field, are there no comparative images of protoplasts/spheroplasts that could be shared?  Perhaps a before-and-after comparison of an intact bacterial cell and a corresponding protoplast/spheroplast shown at the same scale?  Examples of real-world data would help to make the manuscript more appealing.

Specific Comments:

L2: Suggest title change: Factors that affect the enlargement of bacterial protoplasts and spheroplasts

L11+13+18: “Our targets”, “our procedure”, and “Our experimental method” make it seem like this article is more for promotion of the Author’s lab than for an unbiased global review of the topic at hand.  Please change accordingly.

L21: Remove “…based on previous studies and our own experience.”

L28: Change to “…bacteria would display limited genetic variation.”

L30: Change to “…heterologous bacteria greatly contributes to genetic variation.”

L31: Change to “…have been maintained or lost during bacterial evolution?”

L32-33: Change to “Thus far, these remain open questions in the field of molecular biology.”

L47-49: These sound like sentences from a primary research article, not a review article.

L50: Change to “Bacterial cells cannot enlarge in the presence of an intact peptidoglycan sacculus.”

L52: Change to “…because many bacteria can change to …”

L55: Sounds like it is from a primary research article.

L58: Change to “Enlarged bacterial cells have…”

L59: Change to “…but only recently,  a microinjection method has been established for these enlarged cells.”

L66-69: This definition of what constitutes a protoplast vs spheroplast needs to be moved much earlier in the Introduction.  Also, the terms “Gram-positive” and “Gram-negative” should be added (for the purposes of the definition) so that readers are clear on the nature of the different cells being discussed.

L72: Remove “or impossible”.

L92-93: Provide reference(s) for this claim.

L99-100: Provide reference(s) for this claim.

L119: Change to “ions”

L128: What is S-layer?  This needs to be briefly described as it is different from a standard Gram-positive or Gram-negative cell surface.

L210: If novobiocin usage is being mentioned as a way to control bacterial protoplast/spheroplast enlargement, it should be mentioned in a neutral manner.  I.e. “Bacterial protoplast/spheroplast enlargement can be controlled using novobiocin.”

L213: Change to “…DNA attaches to the…”

L216: Change to “…are needed to further understand these…”

L221: Does an increase in bacterial cell size first require DNA replication?  Or does increased cell size promote more DNA replication?  If this is not explicitly known, delete this sentence.

L245: Change to “…inhibitor novobiocin is introduced prior to…”

Author Response

Replies to Reviewer #1 comments:

Thank you so much for your reviewing.

Q1: At the moment, the manuscript reads like more of a promotion for the Author’s lab than for an objective global review of the topic at hand. This is particularly acute in the Abstract (“our targets”, “our procedure”, and “our experimental method”).

A1: I deleted the following, "Our targets for cell enlargement are all bacterial species." I changed "our procedure" to "procedure". I changed "Our experimental method" to "Experiment". I deleted the following, ", based on previous studies and our own experience".

Q2: Are there different ways to make protoplasts/spheroplasts? Mechanical vs enzymatic? What are the benefits/drawbacks? If one way is “better” than another, then describe why.

A2: I changed "Bacterial protoplasts/spheroplasts are produced by lysing the cell wall with lysozyme" to "Bacterial protoplasts/spheroplasts are produced by lysing the cell wall with lysozyme or by penicillin [20]. In the spheroplast incubation method, protoplasts/spheroplasts are produced by lysozyme."

Q3: As presented here, the Author describes various characteristics of protoplasts/spherplasts. However, he also mentions the possibility that these structures are formed in the environment. What would be potential roles/functions for protoplast/spheroplast formation by bacteria in nature? (i.e. what are they used for?)

A3: I added the following, "More works are needed to elucidate functions for bacterial protoplast or spheroplast formation in nature."

Q4: Given the extensive literature and work by the author in this field, are there no comparative images of protoplasts/spheroplasts that could be shared? Perhaps a before-and-after comparison of an intact bacterial cell and a corresponding protoplast/spheroplast shown at the same scale? Examples of real-world data would help to make the manuscript more appealing.

A4: I added one figure as Figure 2.

Specific Comments:

Q5: L2: Suggest title change: Factors that affect the enlargement of bacterial protoplasts and spheroplasts

A5: According to your comment, I changed the title.

Q6: L11+13+18: “Our targets”, “our procedure”, and “Our experimental method” make it seem like this article is more for promotion of the Author’s lab than for an unbiased global review of the topic at hand. Please change accordingly.

A6: Please see A1.

Q7: L21: Remove “…based on previous studies and our own experience.”

A7: Please see A1.

Q8: L28: Change to “…bacteria would display limited genetic variation.”

A8: I changed "not have any" to "display limited".

Q9: L30: Change to “…heterologous bacteria greatly contributes to genetic variation.”

A9: I changed "induces bacterial" to greatly contributes to".

Q10: L31: Change to “…have been maintained or lost during bacterial evolution?”

A10: I changed "accepted or rejected" to "maintained or lost".

Q11: L32-33: Change to “Thus far, these remain open questions in the field of molecular biology.”

A11: I changed "So far, molecular biology has not been able to answer these questions" to "Thus far, these remain open questions in the field of molecular biology".

Q12: L47-49: These sound like sentences from a primary research article, not a review article.

A12: I deleted the following, "We aimed to overcome this problem and make a huge bacterial cell capable of accepting the DNA from a complete bacterial genome through microinjection. The aim goal of our research is to generate a bacterium with a heterologous genome."

Q13: L50: Change to “Bacterial cells cannot enlarge in the presence of an intact peptidoglycan sacculus.”

A13: According to your comment, I changed.

Q14: L52: Change to “…because many bacteria can change to …”

A14: I changed "most" to "many".

Q15: L55: Sounds like it is from a primary research article.

A15: I changed "First, we used the spheroplast incubation method to enlarge bacterial cells" to "The spheroplast incubation method is used to enlarge bacterial cells".

Q16: L58: Change to “Enlarged bacterial cells have…”

A16: According to your comment, I changed.

Q17: L59: Change to “…but only recently, a microinjection method has been established for these enlarged cells.”

A17: According to your comment, I changed.

Q18: L66-69: This definition of what constitutes a protoplast vs spheroplast needs to be moved much earlier in the Introduction. Also, the terms “Gram-positive” and “Gram-negative” should be added (for the purposes of the definition) so that readers are clear on the nature of the different cells being discussed.

A18: I moved "In this review, the bacterial cells lacking cell wall with an outer membrane and a plasma membrane are called spheroplasts, and those without an outer membrane are called protoplasts" and added "Thus, protoplasts and spheroplasts are produced from gram-positive and gram-negative bacteria, respectively."

Q19: L72: Remove “or impossible”.

A19: I deleted "or impossible".

Q20: L92-93: Provide reference(s) for this claim.

A20: I added the references [24, 26, 28].

Q21: L99-100: Provide reference(s) for this claim.

A21: I added the references [24, 28].

Q22: L119: Change to “ions”

A22: I changed to "ions".

Q23: L128: What is S-layer? This needs to be briefly described as it is different from a standard Gram-positive or Gram-negative cell surface.

A23: I added the following, "S-layer is two-dimensional protein array that are frequently found on the bacterial and archaeal surface [40]."

Q24: L210: If novobiocin usage is being mentioned as a way to control bacterial protoplast/spheroplast enlargement, it should be mentioned in a neutral manner. I.e. “Bacterial protoplast/spheroplast enlargement can be controlled using novobiocin.”

A24: I changed to "Bacterial protoplast/spheroplast enlargement can be controlled using novobiocin.”

Q25: L213: Change to “…DNA attaches to the…”

A25: According to your comment, I changed.

Q26: L216: Change to “…are needed to further understand these…”

A26: According to your comment, I changed.

Q27: L221: Does an increase in bacterial cell size first require DNA replication? Or does increased cell size promote more DNA replication? If this is not explicitly known, delete this sentence.

A27: I deleted "These results indicate that an increase in bacterial cell size requires DNA replication."

Q28: L245: Change to “…inhibitor novobiocin is introduced prior to…”

A28: According to your comment, I changed.

Reviewer 2 Report

This review summarized the findings on bacterial protoplast and spheroplast. With the following corrections, I think that this review should be accepted for publish.

Major points

(1)Line 93-94

There is no denying that protoplast is stable in an environment that imitates Seawater, but the existence of protoplast and spheroplast in seawater is a leap in logic.

(2) Line 98-100

The author did not show any evidence of this sentence. It is necessary to show the data or refer the suitable papers.

(3) Line 122-124

This sentence is a leap in logic. It should be deleted, or author should explain in more detail.

(4) Line 125-126

Author should refer the suitable papers.

(5) Line 172-173

It is natural that ATP, which is the main energy source, is required for changes in cell morphology as the enlargement of cells are occurred. Not particularly noteworthy.

(6) Line 222

‘Vacuole’ is likely to be confused with plant vacuoles. With that in mind, in some references the name ‘provacuole’ was used. In ‘review’, these points need to be mentioned correctly.

Minor points

(7) Line 81

What is “brakes”? I couldn’t understand this sentence. Is it “breaks”?

(8) Line 227

What is “amid”? I couldn’t understand this sentence.

(9)Line 317

There is a mistake in the author. I strongly recommend checking again for other references.

Author Response

Replies to Reviewer #2 comments:

Thank you so much for your reviewing.

This review summarized the findings on bacterial protoplast and spheroplast. With the following corrections, I think that this review should be accepted for publish.

Major points

Q(1)Line 93-94

There is no denying that protoplast is stable in an environment that imitates Seawater, but the existence of protoplast and spheroplast in seawater is a leap in logic.

A1: I deleted the following, "This suggests that bacterial protoplasts/spheroplasts may exist in the sea. As previously, mentioned, these cells may have not been isolated because they do not grow."

Q(2) Line 98-100

The author did not show any evidence of this sentence. It is necessary to show the data or refer the suitable papers.

A2: I added two references [24, 28].

Q(3) Line 122-124

This sentence is a leap in logic. It should be deleted, or author should explain in more detail.

A3: I deleted the following, ", indicating that magnesium ions reduce the effect of calcium ions on outer membrane fusion".

Q(4) Line 125-126

Author should refer the suitable papers.

A4: I added two references [26, 37].

Q(5) Line 172-173

It is natural that ATP, which is the main energy source, is required for changes in cell morphology as the enlargement of cells are occurred. Not particularly noteworthy.

A5: I deleted the following, "This strongly suggests that E. litoralis spheroplast enlargement may require the ATP generated during respiration or photosynthesis."

Q(6) Line 222

‘Vacuole’ is likely to be confused with plant vacuoles. With that in mind, in some references the name ‘provacuole’ was used. In ‘review’, these points need to be mentioned correctly.

A6: I added the following, "In this review, "vacuole" is used for "provacuole" or "vacuole-like structure" in previous studies [19, 21]."

Minor points

Q(7) Line 81

What is “brakes”? I couldn’t understand this sentence. Is it “breaks”?

A7: I changed to "breaks".

Q(8) Line 227

What is “amid”? I couldn’t understand this sentence.

A8: "Amid" means "among". So, I changed to "among".

Q(9)Line 317

There is a mistake in the author. I strongly recommend checking again for other references.

A9: I changed to the following reference, "Kawai, Y.; Mercier, R.; Mickiewicz, K.; Serafini, A.; de Carvalho, L.P.S.; Errington, J. Crucial role for central carbon metabolism in the bacterial L-form switch and killing by β-lactam antibiotics. Nat. Microbiol. 2019, 4, 1716–1726."